# The Beneficial Role of Physical Exercise on Anthracyclines Induced Cardiotoxicity in Breast Cancer Patients

**DOI:** 10.3390/cancers14092288

**Published:** 2022-05-03

**Authors:** Eliana Tranchita, Arianna Murri, Elisa Grazioli, Claudia Cerulli, Gian Pietro Emerenziani, Roberta Ceci, Daniela Caporossi, Ivan Dimauro, Attilio Parisi

**Affiliations:** 1Laboratory of Physical Exercise and Sport Science, Department of Exercise, Human and Health Sciences, University of Rome Foro Italico, Piazza Lauro de Bosis 15, 00135 Rome, Italy; eliana.tranchita@gmail.com (E.T.); a.murri@studenti.uniroma4.it (A.M.); claudia.cerulli@uniroma4.it (C.C.); attilio.parisi@uniroma4.it (A.P.); 2Department of Experimental and Clinical Medicine, Magna Graecia University, 88100 Catanzaro, Italy; emerenziani@unicz.it; 3Laboratory of Biochemistry and Molecular Biology, Department of Exercise, Human and Health Sciences, University of Rome Foro Italico, 00135 Rome, Italy; roberta.ceci@uniroma4.it; 4Unit of Biology and Genetics of Movement, Department of Movement, Human and Health Sciences, University of Rome Foro Italico, Piazza Lauro de Bosis 15, 00135 Rome, Italy; daniela.caporossi@uniroma4.it (D.C.); ivan.dimauro@uniroma4.it (I.D.)

**Keywords:** anthracycline, physical activity, breast cancer, LVEF, quality of life

## Abstract

**Simple Summary:**

Breast cancer is one of the most common cancers worldwide and the leading cause of cancer death in women. Screening, early diagnosis, and surgical techniques might increase patients’ survival, leading to a rise in the health-related consequences of anti-neoplastic therapies, such as cardiotoxicity following anthracycline treatments. Alongside conventional therapies, physical activity seems to reduce treatment side effects, improving quality of life either after a breast cancer diagnosis or in the early steps post-surgery. This review offers a general framework for the role of anthracycline in the physio-pathological mechanisms of cardiotoxicity and the effect of exercise on cancer treatment side effects. Moreover, we propose the type and the timing of exercise to better assist patients and reduce the pressure on the health care system in breast cancer patients undergoing anthracycline.

**Abstract:**

The increase in breast cancer (BC) survival has determined a growing survivor population that seems to develop several comorbidities and, specifically, treatment-induced cardiovascular disease (CVD), especially those patients treated with anthracyclines. Indeed, it is known that these compounds act through the induction of supraphysiological production of reactive oxygen species (ROS), which appear to be central mediators of numerous direct and indirect cardiac adverse consequences. Evidence suggests that physical exercise (PE) practised before, during or after BC treatments could represent a viable non-pharmacological strategy as it increases heart tolerance against many cardiotoxic agents, and therefore improves several functional, subclinical, and clinical parameters. At molecular level, the cardioprotective effects are mainly associated with an exercise-induced increase of stress response proteins (HSP60 and HSP70) and antioxidant (SOD activity, GSH), as well as a decrease in lipid peroxidation, and pro-apoptotic proteins such as Bax, Bax-to-Bcl-2 ratio. Moreover, this protection can potentially be explained by a preservation of myosin heavy chain (MHC) isoform distribution. Despite this knowledge, it is not clear which type of exercise should be suggested in BC patient undergoing anthracycline treatment. This highlights the lack of special guidelines on how affected patients should be managed more efficiently. This review offers a general framework for the role of anthracyclines in the physio-pathological mechanisms of cardiotoxicity and the potential protective role of PE. Finally, potential exercise-based strategies are discussed on the basis of scientific findings.

## 1. Introduction

The Global Initiative for Cancer Registry Development has estimated 18.1 m new cancer cases and 9.6 m cancer deaths in 2018 [1]. Breast cancer is one of the most common cancers worldwide and one of the leading causes of cancer death in women [2]. Fortunately, the progression of screening, early diagnosis, and surgical techniques has increased the survival and possibility of recovery, despite the high incidence. On the other hand, the increased survival of breast cancer patients has determined a growing population at risk of developing several other comorbidities and, specifically, treatment-induced cardiovascular disease (CVD) [3,4,5]. In particular, studies have reported that breast cancer patients have an increased incidence of CVD compared to healthy controls, especially those treated with anthracycline (i.e., doxorubicin, daunorubicin, epirubicin, idarubicin, and mitoxantrone) [6]. This class of drug is reported to be the most widely prescribed anticancer agent, but the long-term toxic effect of anthracyclines could lead to increased rates of cardiac morbidity and mortality [7]. To date, it is estimated that the incidence of heart failure in patients receiving doxorubicin is 3%, with a mortality rate of over 60% at 2 years [8]. Prognosis can be improved by prevention, early detection, and treatment; however, a specific treatment for anthracycline-induced cardiotoxicity is not yet available. Recent studies have highlighted that regular physical exercise (PE), improving cardiorespiratory, musculoskeletal, and endocrine systems, has an important role in maintaining health and inducing mechanisms able to prevent these diseases [9,10,11,12,13,14,15]. Indeed, PE is considered a consolidated protective factor for the prevention and treatment of the main non-communicable diseases (NCDs), such as cancer and cardiovascular diseases [16]. In oncology, PE is becoming increasingly important among integrative therapies, improving the quality of life (QoL), and relieving the symptoms and side effects of the treatments and of the disease itself [16]. In particular, with regard to women with breast cancer, exercise seems to drastically reduce the risk of relapse [17,18], limits body weight gain [19], improves bone health [20], and increases the life expectancy of patients [21] improving physical and mental well-being [22,23]. The current PE prescription guidelines for cancer patients suggest a moderate-intensity aerobic activity of 150–300 min/week or 75–150 min/week at vigorous intensity, or an equivalent combination of both; resistance training, two or more times a week for the main muscle groups (50–70% of one-repetition maximum-1 RM), and joint mobility, 2–3 times a week [24]. Despite these recommendations, few studies have highlighted the impact of PE on cardiotoxicity mechanisms induced by drugs in breast cancer patients, therefore the need to develop tailored complementary strategies, alongside drug therapy, is still a priority [25,26,27,28,29]. This is even more important in the recent period, in which COVID-19, the infection caused by the severe acute respiratory syndrome coronavirus 2, has generated high pressure on the healthcare system. According to the latest studies, the pandemic has negatively impacted the management of NCDs [30]. The World Health Organization (WHO) conducted a rapid assessment survey of service delivery for NCDs during the COVID-19 pandemic and the evidence reported that many people living with NCDs are no longer receiving appropriate treatment or access to medicines [31]. The treatment of cardiovascular and oncologic diseases seems to have been affected mostly by COVID-19. Considering this dramatic situation, the need for co-adjuvant and effective therapies capable of positively managing oncologic symptoms and treatment-induced side effects is even more pressing. Despite the strong evidence, it is not clear which exercise should be suggested in breast cancer patients undergoing anthracycline, and which mechanisms are behind the protective role of PE in those patients. This review offers a general framework about the role of anthracycline in the physio-pathological mechanisms of cardiotoxicity, and the role of PE on those mechanisms able to reduce the anthracycline side effects in cancer. Then, we mean to provide a strategy to alleviate the pressure on the healthcare system, as well as guidelines about the type and the timing of exercise in breast cancer patients undergoing anthracycline.

## 2. Physio-Pathological Mechanisms of Cardiotoxicity Induced by Anthracyclines

Anthracyclines represent a cytotoxic chemotherapeutic drug often used to treat different types of cancers such as breast cancer, lymphoma, sarcoma, and others [32]. It is used either alone or in combination with other drugs, surgery, and/or radiation [32]. The role of doxorubicin and other anthracycline drugs has been fully investigated during the last decades, but the supposed molecular mechanisms for cell killing are still controversial. Several models have been proposed for doxorubicin-mediated cell death, including topoisomerase II poisoning, DNA adduct formation, oxidative stress, and ceramide overproduction [33,34,35]. The biological activity of this anticancer drug is due to its capacity to avidly bind to DNA by intercalation, causing inhibition of DNA synthesis. The strand separation that occurs during intercalation unwinds the double helix and produces DNA supercoils, resulting in increased torsional stress. This event can affect the structure and dynamics of nucleosomes, the repeating unit of chromatin composed of DNA, wrapped around octameric histone cores. Indeed, this torsion-induced nucleosome destabilization is emerging as a significant molecular mechanism for the action of doxorubicin and related anthracycline drugs [36,37,38]. All of these different models of doxorubicin action are consistent with the broad spectrum of activity of the drug in cancer treatment. Although it is likely to be multifactorial, the exact mechanism of anthracycline-induced cardiotoxicity remains unclear. The most widely accepted hypothesis suggests an interaction of anthracyclines with the redox cycle, resulting in DNA damage related to reactive oxygen species (ROS) production.

In 1980 three distinct types of anthracycline-induced cardiotoxicity were identified, as is summarized in Figure 1 [25]:○acute cardiotoxicity may occur after a single dose, or a single course, with the onset of symptoms within 14 days from the end of treatment, and it is usually reversible.○early-onset chronic cardiotoxicity, occurring within 1 year, presenting as dilated-hypokinetic cardiomyopathy, with progressive evolution toward heart failure. ○late-onset chronic cardiotoxicity, developing many years after the end of anthracycline therapy.

It is well known that cardiotoxicity is probably a continuous phenomenon, starting at the level of the cardiomyocyte, followed by progressive functional decline and leading to overt heart failure at the end. As already suggested by Cardinale et al. [39] in their recent work, we are probably observing the evolving stages of the same phenomenon and not three distinct diseases. Therefore, it is important to understand that the occurrences of heart failure are dose- and schedule-dependent. The diagnosis of anthracycline-induced cardiotoxicity was made based on heart failure symptoms and recently was also based on left ventricular ejection fraction (LVEF) seen at echocardiography. Cardinale et al. [39] proposed as a parameter to define heart failure an LVEF absolute decrease which is 10% higher than baseline, to a decline of LVEF < 50%. Similar values were proposed by Plana et al. [40] that define LVEF decreasing > 10% points, with a final value of <53% as the limit value to diagnose cardiotoxicity-related heart failure. International guidelines, although stressing the importance of serial echocardiography in subjects receiving chemotherapeutic drugs, do not provide an accurate indication of timing, frequency, modalities, and long-term schedule. Consequently, there is the risk of late diagnosis of heart failure, when it may be irreversible [25,41]. At the same time, the sign we could recognize in echocardiography represents a tardive phase of cardiotoxicity, when the injuries of cardiomyocytes have begun. Recent studies have suggested biochemical markers, such as Troponin and natriuretic peptides, as possible indicators of cardiotoxicity at an early preclinical stage, before symptoms of heart failure occur, and before an asymptomatic drop in LVEF [42,43]. Finally, the latest echocardiographic methods have highlighted sensitive parameters for the early identification of cardiotoxicity. For example, tissue Doppler and Strain imaging techniques can identify early changes in cardiac function, before LVEF falls [39]. The most common clinical presentation of cardiotoxicity is dilatation-hypokinetic cardiomyopathy leading to heart failure, although this is only the latest phase of cardiotoxicity. We know that the cardiovascular system has several targets that can be injured by anthracycline. Firstly, anthracycline can directly damage cardiomyocytes or induce inflammation of the pericardium. Secondly, chemotherapeutic agents may affect the coagulation system predisposing to thromboembolic events and consequent cardiovascular ischemia. These cardiovascular conditions may induce several pathologies such as acute myocarditis, pericarditis, ventricular and supraventricular arrhythmia, as well as myocardial ischemia and cardiogenic shock [44].
Figure 1Anthracycline-induced cardiotoxicity mechanisms. Abbreviation: LVEF, left ventricular ejection fraction. CM, Cardiomyocytes.
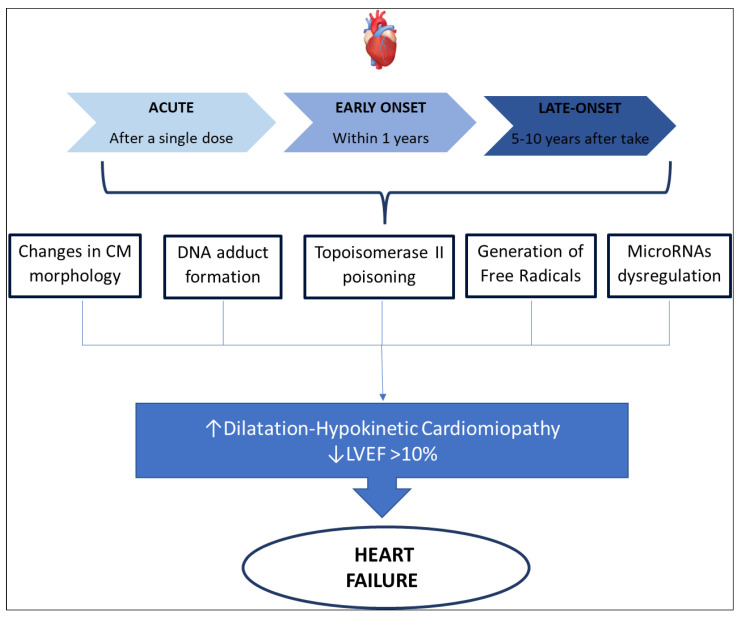


## 3. Molecular Landscape of Exercise Benefits in Anthracycline-Induced Cardiotoxicity

Physical exercise, both aerobic and strength, could represent a viable non-pharmacological treatment, as it increases the “cardioprotective” function by inducing a series of biomolecular mechanisms regulating mainly the redox-homeostasis and cell survival [13,26,45,46]. However, to better understand how exercise can provide cardiac protection against anthracycline-induced cardiotoxicity (AIC), it is important to keep in mind the major molecular mechanisms by which this class of drugs induces cardiotoxicity. Several studies have described the mechanism underlying the antitumor function of these compounds [47,48,49]. Among the proposed mechanisms of cardiac injury, anthracyclines-induced generation of reactive oxygen species (ROS) appears to be the central mediator of numerous direct and indirect cardiac adverse consequences [50,51,52]. In particular, it is known that anthracyclines carry out their antitumoral effects through direct inhibition of topoisomerase II in cancer cells (proliferating cells), more specifically the 2α isoform, impeding DNA transcription, and replication. This mechanism cannot explain the effect of anthracycline on non-dividing cells such as cardiomyocytes. The only topoisomerase II inside cardiomyocytes is expressed in mitochondria (Topoisomerase 2β) [53]. Supporting this hypothesis, Zhang et al. [54] demonstrated that anticancer treatment based on doxorubicin induces important changes in the gene expression of both mitochondrial structure and metabolism in cardiomyocytes expressing Topoisomerase 2β. Moreover, treatment with anthracyclines may decrease the transcription of two genes (encoding for PGC-1α and PGC-1β), reducing the mitochondrial activities in cardiomyocytes [55]. This reduction induces a supraphysiological production of ROS [56,57,58], therefore cardiomyocytes exposed to oxidative stress activate the tumor suppressor protein p53 and a multiplicity of signaling pathways such as p38 mitogen-activated protein kinase (MAPK) and c-Jun N-terminal kinases (JNKs), leading cells to programmed cell death, also known as “myofilament apoptosis”, deriving from suppression of myofilament protein synthesis and alteration of cardio-myocytes’ morphology [59,60,61]. These changes are induced by the excessive release of calcium, combined with the inhibition of its reuptake into the sarcoplasmic reticulum. Therefore, an overload of intra-cytoplasmatic calcium occurs with subsequent systolic/diastolic dysfunction. At the molecular level, this dysfunction is induced by the release of cytochrome c, a pro-apoptotic factor, and by activating the cysteine protease calpain, which cleaves and regulates structural myofibrillar proteins [62,63,64]. Differently, the suppression of myofilament protein synthesis is generated by the depletion of cardiac progenitor cells (CPC) and the down-regulation of GATA-4, which is a CPC transcriptional factor essential for postnatal cardiomyocyte survival. This leads to the inhibition of sarcomere protein synthesis, the increase in senescent cardiac cells, and ventricular dysfunction [65,66,67] (“Suppression of Myofilament Protein Synthesis and Ultrastructural Changes to Myocytes”). The alteration in cardiac energy metabolism is produced by the reduction of ATP and phosphocreatine levels and AMP-activated protein kinase (AMPK) activity. These events lead to a lack of acetyl-CoA carboxylase, and thereby to an impairment of fatty acid oxidation [68,69] (“Alterations in Cardiac Energy Metabolism”). 

The first study evaluating the role of exercise in AIC in a mouse model was published in 1979 [70]. The authors tested the effect of an acute exercise stress (swim for 30 min) upon the acute administration of doxorubicin (DOXO, 18 or 23 mg/kg). Thirty days later, the survival of the exercise group was not different, compared to the control group (sedentary animals) demonstrating that exercise stress does not increase DOXO toxicity. A few years later, Kanter et al. [71], and then Ji and Mitchell [72], investigated the role of exercise to prevent ROS formation and reduce cardiac oxidative stress induced by anthracycline. In the earlier study [71], mice receiving doxorubicin (4 mg/kg; 2 days/week for 7 weeks) were divided into two groups: mice that underwent a swim training program (60 min/day; 5 days/week for 21 weeks), and sedentary mice. After 21 weeks, the trained mice had elevated levels of blood catalase (CAT), liver CAT, superoxide dismutase (SOD), and glutathione peroxidase (GPx). Moreover, the level of cardiotoxicity was significantly greater in the sedentary group than in the trained group [71]. In 1994, Ji and Mitchell [72] investigated the effect of DOXO administration (bolus 4 mg/kg; twice, 24 and 1 h prior to killing animals) on cardiomyocyte mitochondria in rats at rest and after an acute bout of exhaustive, graded running on the treadmill (≈30 min). Results showed that DOXO could interfere with normal heart mitochondrial function both at rest and during heavy exercise. The mitochondrial respiratory control index was decreased with DOXO administration, but the reduction was due to an increase in state 4 rather than a decrease in state 3 (ADP-stimulated) respiration. Following these studies, many others have attempted to understand the role of exercise in the prevention of ROS formation in CIC. Particularly, Ascensao et al. [73,74] demonstrated that a period of swimming endurance training in mice (60 min/day; 5 days/week; 14 weeks), followed after 24 h by a treatment with a bolus of DOXO (20 mg/kg), increases the level of total and reduced glutathione, increases the activity of CAT and SOD, the expression of HSP60 and HSP70, and reduces the rise of plasma cardiac troponin I (cTnI) compared with the sedentary mice as controls. These changes result in an improvement of the mitochondrial and cell defense systems and a reduction of cellular oxidative stress. Moreover, Werner et al. [75] reported that voluntary exercise for 21 days (mean distance 5.100 ± 800 m/day) could prevent DOXO-induced cardiomyopathy (22.5 mg/kg for 24 h) through the modulation of apoptosis, reducing mitochondrial levels of protein carbonyl groups, malondialdehyde, Bax, and caspase-3 activity. On the same wavelength, Chicco et al. [76,77] highlighted firstly that either hearts from rats left free to wheel-running for 8 weeks and then perfused with buffer containing DOXO (10 μM) for 60 min, or hearts from rats trained on a motorized treadmill for 12 weeks (20–60 min/day, 15–27 m/min), which had received DOXO (15 mg/kg) before being killed, had an attenuation of DOXO-induced lipid peroxidation and higher levels of HSP72. Secondly, they proposed low-intensity exercise training during the course of DOXO treatment to reduce the activation of apoptosis and cardiac dysfunction [78]. At the molecular level, the authors found that rats trained on a motorized treadmill (20 min/day, 15 m/min, 5 days/week) for 2 weeks, which received DOXO (2.5 mg/kg, 3 days/week) during the same 2-week period, had a significant increase in HSP70 and GPx in the cardiac tissue. These results were further confirmed by other subsequent studies [79,80,81]. Particularly, Shirinbayan et al. (2012) demonstrated that 3 weeks of regular aerobic treadmill running (25–39 min/day, 15–17 m/min, 5 days/week) before DOXO administration (10–20 mg/kg) significantly increases cardioprotective markers such as SOD activity and HSP70, and decreases malondialdehyde, creatine kinase, and creatine phospho-kinase [81].

One of the most relevant side effects of AIC is the reduction in CPC differentiation and proliferation. Two independent studies demonstrated that exercise significantly increases the number of CPC in hearts [82] and the level of mRNA GATA-4 [83], an important transcriptional factor for cardiomyocyte survival. According to these two studies, aerobic exercise prevents ROS formation, limits oxidative stress, and decreases the suppression of myofilament protein synthesis. Multiple experiments by Hydock et al. [84,85,86,87] report that different types of regular physical activity (10 weeks), treadmill or voluntary wheel running, in rats before and during DOXO treatment (10–15 mg/kg) maintain a higher level of α-myosin heavy chain (MHC) isoform in cardiac cells than in the sedentary control rats. This is one of the various cardioprotective mechanisms of exercise against the effects of anthracycline, since an increase in β-MHC isoform is associated with heart failure. A similar result was also found by Pfannenstiel et al. [88]. In particular, the authors reported that 12 weeks of resistance training in rats before DOXO treatment (12.5 mg/kg) preserves cardiac function and attenuates the α-to β-MHC shift that occurs with DOXO treatment [89]. Finally, a further positive effect of exercise on the heart muscle could be the activation of AMP-activated protein kinase (AMPK). Indeed, one of the best-characterized downstream targets of AMPK is acetyl-CoA carboxylase. Its phosphorylation inhibits malonyl-CoA synthesis, enhancing carnitine palmitoyl-transferase I activity and free fatty acid oxidation [90]. However, this beneficial effect of exercise has not yet been tested after doxorubicin administration in animals.

In conclusion, the aforementioned preclinical studies suggest that the main mechanisms used to protect the body from CIC are (Figure 2):the enhancement in the production/activity of CAT, SOD, GPx, HSP60, HSP70, total and reduced glutathione, as well as the decrease in pro-apoptotic signaling from Bax, caspase-3, and p53 expression against reactive oxygen species (ROS) production;the stimulation of the proliferation and mobilization of CPC and the expression of GATA-4 mRNA for re-establishing the ultrastructure of cardiac microfilaments and preventing calcium overload;the increase in AMPK levels, resulting in improved cardiac metabolism so that both ATP and creatine phosphokinase levels increase;the support of cell survival by preventing the high levels of autophagy/lysosomal signaling (i.e., LC3II/LC3I, ATG12, ATG4, ATG5, and ATG7 proteins) usually induced by doxorubicin [89].

Taken together, these pre-clinical studies suggest that exercise (aerobic and resistance) is an appealing and recognized, supportive therapy to prevent AIC. As observed in all studies, the different cardioprotective mechanism of exercise seems to depend on its characteristics (i.e., type, duration, intensity, and frequency) and animal model.

Currently, the most appropriate and efficacious exercise prescription for preventing and/or treating anthracycline-induced cardiotoxicity is not known. Further human clinical studies will be necessary to elucidate the molecular mechanisms, including the role of circulating mRNAs, underlying the cardioprotective properties of exercise training before, during, and after anthracyclines exposure, and specifically to define the nature and magnitude of the effect of exercise on chemotherapy-induced cardiotoxicity in cancer patients.
Figure 2Schematic representation of the mechanisms underlying anthracycline-induced cardiotoxicity and their modulation through physical exercise in breast cancer. Abbreviation: ROS, reactive oxygen species.
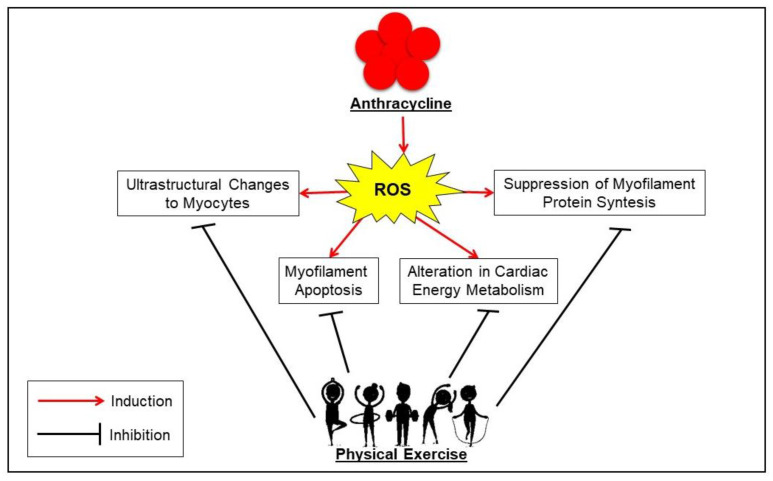


## 4. Search Method

The purpose of this article is to review a series of studies highlighting the possible positive and protective role of physical exercise on cardiotoxicity induced by cancer-related treatment, in particular anthracycline, in breast cancer patients. A literature search was conducted by two authors using three databases (PubMed, Google Scholar, and Scopus). The search was limited to peer-reviewed journals written in English. Keywords used in searches included «aerobic» «strength» «combined» «training» «physical activity» «doxorubicin» «cancer» «treatment» «cardioprotection» «FEVS» «GLS» «Troponin» «NT-proBNP» « VO_2_max» «CRF» «RCT». Inclusion criteria consisted of studies regarding the effects of exercise on cardiotoxicity in breast cancer patients. A manual search of the reference lists in the studies found in the computerized search was also conducted. Once the duplicates were removed, the total search yielded 18 studies that matched the above-mentioned criteria. The full search was manually retrieved (Figure 3).

## 5. Cardioprotective Role of Physical Exercise in Breast Cancer Patients Undergoing Anthracycline

In all tissues, PE improves the homeostasis of various macromolecules (DNA, RNA, and proteins) involved in the response to physiological or pathological stress [91,92,93,94,95,96,97]. Indeed, the induction of adaptive mechanisms elicited by regular exercise at systemic or at tissue-specific levels has been shown to delay the onset and progression of several diseases and aging-related biomarkers [98,99,100,101,102]. To date, a cardio-oncological rehabilitation program based on PE seems to be essential for all patients undergoing cardiotoxic chemotherapy regimens such as doxorubicin [13,14,93]. Unfortunately, few studies have been performed on humans and therefore it is still a priority to identify the optimal characteristics of PE and the more appropriate timing to start a tailored training program. Evidence has reported that, in a healthy population, PE has a positive impact on echocardiographic parameters such as LVEF, Global Longitudinal Strain (GLS), end-diastolic and end-systolic volume, E/A, E/e ’, dP/dt_max_ [103,104,105], and decreases the cardiac Troponins level [106]. As reported by Hamasaky et al. [107], tailored exercise programs seem to induce an increase in NT-proBNP and NP during exercise, which return to the basal levels after 72 h, both in a patient with cardiac dysfunctions and in a healthy population. According to this evidence, it seems clear that PE could have a pivotal role in the management of risk factors related to CVD, especially in women with breast cancer undergoing anthracyclines treatment. Interestingly, combined training (aerobic and strength training) can reduce the “Framingham risk score”, which is a predictor of the ability to develop CVD in the next 10 years [108]. Moreover, a growing number of scientific evidences has been showing that exercise, in particular aerobic training, seems to counteract oxidative stress in those patients treated with doxorubicin, which seems one of the main causes of the onset of cardiac dysfunction [109,110,111]. Aerobic training, through the positive impact on the antioxidant systems, seems to reveal a protective role on cardiomyocytes’ health, decreasing the reactive oxygen species (ROS) levels responsible for pathological mechanisms in the cascade. Exercise decreases pro-apoptotic mediators’ levels by counteracting stress-induced cellular apoptosis, reduces the inhibition of protein synthesis, and improves the proliferation and energy metabolism of cardiomyocytes [26,46,112]. To better understand the latest effect of training, according to the time of administration, on cardiopulmonary and cardiac functions, as well as on biomarkers related to cardiotoxicity induced by treatments on breast cancer patients related with anthracycline, in particular DOXO, this review focused on studies published between 2007 and 2021. A summary of the reviewed studies is presented in Table 1.

### 5.1. Aerobic Training and Cardiopulmonary Function

Aerobic exercise, also known as endurance activities, cardio or cardio-respiratory exercise, is physical exercise of low or high intensity that depends primarily on the aerobic energy-generating process. It is well known that this type of training produces favorable changes in myocardial function, symptoms and functional capacity, and increases hospitalization-free life span and probably survival in patients affected by cardiovascular diseases [113] The first studies on PE and cardiotoxicity focused on the improvement in cardiopulmonary function and VO_2_max after aerobic training, in women affected by breast cancer during adjuvant therapies. Participants of the study of Courneya et al. [114] exercised for the duration of the entire chemotherapy and 17 weeks after treatments. Patients were randomly assigned to an aerobic training group (from 60 to 80% of VO_2_max), to a strength training group (60% to 70% of their estimated one repetition maximum), and to a usual care group (no exercise). The results revealed that both trainings were feasible and well tolerated by the women. Data showed an improvement in aerobic fitness, QoL, fatigue, depression, and anxiety in the aerobic group compared to the usual care group, and a positive change in lean body mass, QoL, self-esteem, and depression after resistance training compared to the usual care group. The results suggest that aerobic training seems to have a major cardioprotective role in increasing peak oxygen consumption, which is one of the strongest predictors of survival in patients with chronic congestive heart failure [115]. The relation between aerobic training and cardiopulmonary function in breast cancer undergoing anthracycline was deeply investigated by Horsnby et al. [116]. They enrolled 20 patients with stage IIB-IIIC operable breast cancer, undergoing doxorubicin plus cyclophosphamide, and assigned them to the anaerobic group, that performed three supervised sessions per week at 55–100% of patients’ exercise capacity for 12 weeks, using a cycle ergometer, or to a no-exercise group. According to the previous study by Courneya et al. [114], the results showed an increase in the peak oxygen consumption and oxygen pulse; comparing the exercise with the no-exercise group, no significant results were found in the echocardiography analysis. It seems that oxygen pulse during exercise could be a predictor of coronary heart disease and all-cause death, providing an additional means for defining the prognosis for patients with CHD [117]. Most recently, Moller et al. [118], investigated the effect of supervised and non-supervised training on cardiorespiratory fitness (CRF/VO_2_peak) assessed during 6 weeks, after 12 weeks of the training, and after 6 months of follow-up, in physically inactive BC patients undergoing adjuvant chemotherapy that included a sequential anthracycline. The supervised protocol consisted in a first part, lasting 6 weeks, that included three sessions per week of high-low-intensity combined training and one session per week of restorative exercises including massage; the second part, lasting 6 weeks, included supervised floorball games, dance, and circuit training, for a total of 12 weeks of training. The non-supervised protocol consisted of a home-based pedometer intervention, health counseling, and symptom guidance, where the goal was to achieve 150 min of moderate-to-vigorous PA per week. Results of the latter study showed that no difference in VO_2_peak was observed between the two groups after 12 weeks of training, instead, a decline of this parameter was observed in both groups after the onset of chemotherapy. The VO_2_peak was fully restored in both groups after 6 months of follow-up, suggesting a long-term effect of training on cardiorespiratory fitness and revealing a decline in cardiovascular risks, in both groups. To our knowledge, the latter study is one of the first that focus on a longitudinal approach to physically active screened survivors. According to these studies, 12–17 weeks of aerobic training, performed during treatments, is able to prevent and reduce the side effect of anthracycline in BC patients, improving cardiopulmonary function. These positive effects seem to persist even after the exercise protocol.

### 5.2. High-Intensity Interval Training and Cardiopulmonary Function

High Intensity Interval Training (HIIT) can be broadly defined as repeated bouts of short or moderate duration exercise (i.e., 10 s to 5 min) completed at an intensity that is greater than the anaerobic threshold [119]. Exercise bouts are separated by brief periods of low-intensity work or inactivity that allow a partial but often not a full recovery. This type of training is well-tolerated, even in elderly people, and improves VO2peak and diastolic function, inducing important implications for health outcomes in patients affected by cardiovascular diseases [120]. A study investigated the effect of HIIT on VO_2_max and Peak Power Output (PPO) compared to a no-exercise group, in patients undergoing anthracycline-based chemotherapy [121]. The exercise intensity was prescribed by considering PPO, and was evaluated on a cycle ergometer. Participants exercised three times a week for 8 weeks on a stationary bike. Each HIIT training session included seven sets composed of a 1-min interval performed at 90% PPO, followed by a 2-min interval performed at 10% PPO. This intervention was feasible, and no side effect was reported during and after the training. Data on VO_2_max and PPO reported no significant increase in the HIIT group, but both parameters significantly decline in the control group; additionally, group interactions were not found. The non-significant differences might be explained considering the low volume of exercise, which might counteract the decline of VO_2_max and PPO induced by anthracycline, in particular during DOXO treatment. After this preliminary data, the same authors evaluated the effect of the same protocol on vascular endothelial function and vascular wall thickness, which are negatively affected by anthracycline toxicity, increasing the risk of CVD development. After 8 weeks of intervention, the HIIT group showed a significant increase in brachial artery flow-mediated dilation (baFMD) and maintenance of carotid intima-media thickness (cIMT), while the control group showed a significant reduction in baFMD and an increase in cIMT. Results suggested that this type of HIIT protocol in BC patients undergoing anthracycline seems to have a cardioprotective role, improving vascular endothelial function and maintaining vascular wall thickness [121]. Moreover, Lee et al. [122] studied the protective effect of the same HIIT protocol on enzyme levels of matrix metalloproteinases (MMP-2, MMP-9), which are regulators of the extracellular matrix structure within the vascular system, and seem to be overexpressed by Anthracycline, increasing the overall cardiovascular risk. After the HIIT training, results showed a significant decrease in MMP-9, whereas the level of this marker was maintained in the control group. Both groups evidenced an increase in the MPP-2 enzyme. Data suggest that this type of exercise has a partial cardioprotective role, decreasing MMP-9 levels, which are correlated with blood pressure. However, more studies are needed to understand the most effective volume, intensity, and timing of HIIT administration. To increase knowledge of the cardio-protective role of exercise in breast cancer undergoing anthracycline, Ma et al. [111] pointed out that aerobic training performed during treatment might improve both pulmonary and cardiac functions. All patients received four cycles of anthracycline chemotherapy; before the first cycle they were randomly assigned to a control group, no-exercise group, or to an aerobic training group, where they performed 50 min of activity, three times per week, for 16 weeks at 90–95% of patients’ maximal heart rate (HR_max_). Results showed that a high-intensity aerobic intervention not only improves VO_2_max and consequently lung functions, but can also improve heart functions, increasing LVEF. On the other hand, the control group showed a significant decrease in the same parameters, as well as in E/A and DT intervals, revealing the myocardial damage induced by anthracycline. The results of the latter study highlight that aerobic exercise could improve heart function attenuating the damage of chemotherapy drugs. It seems that this type of training, performed during anthracycline treatments, can improve both cardiac and pulmonary functions, protecting BC patients from cardiotoxicity. However, if HIIT protocol seems feasible for these patients, further studies are needed to support these results.

### 5.3. Combined Exercise and Home-Based Training

Combined or Concurrent Training is defined as the combination of resistance and endurance training. The combination of these two types of exercise in a training program leads to superior adaptations in health-related and body function variables, regardless of age or sex, including increases and/or improvements in basal metabolic rates, insulin sensitivity, glucose/lipids metabolism, lipidemic profile and body composition, while both muscular hypertrophy/strength/power and endurance capacities are increased [123]. In fact, this type of training seems one of the most effective in reducing cardiovascular risk in both men and women [124]. Kirkham et al. [125] tried to assess the association between clinical indices of cardiovascular autonomic function and exercise training through a longitudinal study on breast cancer patients undergoing treatments. Participants were encouraged to follow a protocol that combined supervised and home-based exercise; no strict expectations were established. Patients undergoing treatment started with training consisting of 20–30 min of aerobic training (50–75% of their heart rate reserve) and whole-body resistance exercise (50–75% 1 RM), three times per week combined with one–two sessions per week of a home-based aerobic protocol. After the treatments, the women were encouraged to perform two supervised and three home-based sessions per week for 10 weeks, and then, patients performed one supervised and four home-based sessions per week for 10 weeks. Seventy-three patients were enrolled, and data were collected before, and after the treatment, 10 weeks, and 5–6 months after the end of the protocol. Data showed that during chemotherapy HR rest increased, and SBP rest and DBP rest decreased, suggesting the negative effect of therapy during the time. Moreover, results emphasized that approximately one-third of these patients suffered from tachycardia and diastolic hypotension, which should be regularly monitored by professional trainers. However, exercise training improved aerobic fitness, evaluated through the estimated peak volume of oxygen consumption by a modified Balke protocol, in those patients that attended at least two out of three supervised sessions per week, mitigating the negative side effect related to treatment. Another longitudinal study investigated the cardioprotective role of exercise during and after treatment in breast cancer patients [126]. 603 patients were enrolled, and echocardiographic outcomes and self-reported physical activity levels, using the Godin Leisure-Time Exercise Questionnaire, were assessed on the first day of cancer therapy, every 4–6 weeks during cancer therapy, and every year after the end of therapy. Results showed that, despite the level of self-reported PA being low immediately before the beginning of treatment, exercise increased significantly in the 1–2 years after breast cancer therapy. The multivariate analysis revealed a moderate positive linear relation between self-reported physical activity and LVEF over an extended follow-up time, suggesting the positive effect of continuous training on echocardiography-derived measures. The strength of these studies is the possibility of generaling the results to the whole breast cancer population, including a wide range of ages, multiple ethnicities, comorbid conditions, and common treatments, increasing the body of evidence in this area. However, interventional studies are needed to deeply understand the cardioprotective role of exercise and to assess new markers able to predict cardiotoxicity induced by anthracycline. Early detection and classification of cardiotoxicity, through feasible clinical equipment, may be crucial to developing targeted intervention and preventing mortality. Howden et al. [127] demonstrated that a well-structured exercise might influence the VO_2_peack, which is correlated with heart failure and mortality [128]. The authors hypothesize that cardiac reserve could be a more sensitive marker for early cardiac dysfunction than LEFV. From this perspective, women before the beginning of the treatment chose to participate in the exercise group or the control group. The exercise group attended 12 weeks of combined training, 30 min of aerobic and 30 min of strength exercise, twice a week, and 30–60 min home-based aerobic exercise sessions, once a week, while the control group received the standard medical care. Data showed a significant reduction of LVEF and a significant increase in Troponin I in the exercise group, but no difference was observed between groups. Although the VO_2_peak did not increase in both groups, the exercise group showed a significantly lower VO_2_ decrease after training than in the control group. The author suggested that this decline can be partially attributed to abnormalities in microvascular functions and to consequent decrease in oxygen diffusive conductance within the skeletal muscle. Moreover, contrary to expectations, exercise did not influence the cardiac reserve, suggesting that more studies are needed to understand the potential role of this parameter for early cardiac dysfunction, as well as the proper intensity and timing of exercise. The cardioprotective role of combined training in BC cancer treated with anthracycline, both supervised or performed at home, is still controversial. It seems that at least 4 months of training can induce positive effect on LEFV, but does not influence cardiac reserve.

### 5.4. Exercise and Cardiac Function in Breast Cancer Patients

Kirkham [129] evaluated if training during four cycles of treatment can affect cardiac function and hemodynamics. The protocol was performed three times per week and consisted of 20–30 min of aerobic exercise (on the treadmill, elliptical, or cycle ergometer) at 50–75% of age-predicted heart rate reserve, and included whole-body resistance exercises. Data from this study confirmed the results shown by Howden et al. [127], where no impact was detected in resting diastolic and systolic cardiac function, including GLS and LVEF after exercise, and a small decrease in VO_2_peak was evidenced. Cardiac output increase was higher in the control group, probably due to an elevated heart rest rate, that in the exercise group, where it was less evident. These data, alongside the lack of adverse events, supported the positive effect of exercise on this population. In addition, exercise seems to positively impact resting hemodynamics, attenuating the hematocrit reduction. The hemodynamic improvement, in this study, was supported by the attenuate systemic vascular resistance (SVR) reduction detected in the exercise group compared with the usual care. These results suggested that exercise could attenuate anthracycline side effects in hemodynamics, despite the decrease in VO_2_peak. Kirkham et al. [130] evaluated the relation between VO_2_peak, LV, and Myocardial T1, which is an index for myocardial fibrosis. They enrolled BC women that did not receive treatments, BC women that received anthracycline treatment, and healthy women; the level of physical activity was evaluated through the Godin Leisure-Time Exercise Questionnaire, and a Maximal Exercise Test on a cycle ergometer was assessed. All participants were divided into “fit” or “unfit” according to their VO_2_peak. Data evidenced that LV end-diastolic volumes and ejection fractions were similar in all groups. BC women undergoing anthracycline showed a decreased VO_2_peak compared to the healthy control group and BC women that do not receive treatments. Moreover, BC women who did not receive treatment reported the same Myocardial T1 and VO_2_peak values as healthy women. Interestingly, all fit women reported a similar T1, suggesting the positive impact of physical activity on myocardial fibrosis, even during anthracycline treatment. Nagy et al. [131] evidenced the relationship between an active lifestyle and diastolic function (E/A) in breast cancer survivors that received anthracycline treatment. 55 participants were divided according to their physical activity level. The diastolic function was evaluated before the beginning of the treatment, during the therapy, 1 year, 2 years, and 5 years from the first anthracycline treatment. Both groups showed a decreased E/A during treatment, but in the active group, who performed at least 30 min of intensive exercise for 4–5 days, heart failure symptoms related to treatments were less frequent. The study supports the evidence that an active lifestyle can attenuate the side effect of cancer treatments.

### 5.5. Exercise and Biochemical Markers of Cardiotoxicity

To understand the effect of exercise on subclinical and biochemical markers related to cardiotoxicity, Kirkham et al. [110] evaluated the role of a single treadmill session on cardio-protection in breast cancer patients. Participants were randomly divided into two groups, one performing the usual care, and the other one performing 30 min of treadmill bout (70% of their Heart Rate Reserve) approximately 24 h before the doxorubicin treatment. Clinical parameters and circulating biomarkers were evaluated 0–14 days before the first doxorubicin treatment and 24–48 h after. NT-proBNP, an early circulating biomarker that is prognostic for anthracycline-related cardiac injury and events, showed an increase in both groups, but the group involved in exercises seemed to attenuate the increase of this biomarker compared to the control group. On the other hand, no significant interaction or changes over time were observed at the cTnT level. Data also showed an increase in the strain rate (SR) and LVEF in the training group. The combination of these effects could indicate an exercise-induced increase in systolic function, perhaps partially due to a reduced afterload, but future studies with larger sample sizes are required. The same author evaluated the effect of four bouts of aerobic training, already described [110], on LV longitudinal strain, twist, cardiac Troponin T, NT-proBNP, and cTnT [28]. Twenty-four women were randomly assigned to a usual care group, or to a supervised training group, that performed the training approximately 24 h before each of four doxorubicin treatments. Data showed a positive effect of this protocol on cardiovascular hemodynamics, such as an increase in cardiac output and resting HR, a decrease in systemic vascular resistance and arterial pressure, as well as body weight. Again, no interaction between the groups was found in circulating biomarkers, but a significant increase in cTnT and NT-proBNP was found in both groups after treatment. Therefore, aerobic exercise intervention might improve several cardiovascular parameters, despite that the mechanisms behind this result are still not clear. Moreover, because NT-proBNP was not affected by training, the author suggests assessment of this biomarker to standardize cardiovascular outcomes across time points [28]. Most recently, Heinze-Milne et al. [132] studied the potential effect of 12 weeks of aerobic exercise on the systemic level of inflammatory cytokines in breast and hematological cancer patients receiving anthracycline treatment. Forty-nine women (*n* = 28 breast, *n* = 21 hematological) performed aerobic exercise at 35–85% Heart Rate Reserve (HRR) on the treadmill for 20–45 min per session under the supervision of research staff. The study showed the feasibility of intervention achieving 73% of adherence with no exercise-related adverse events. Data on cardiopulmonary fitness, physical activity levels, quality of life, fatigue, and inflammatory cytokines did not change throughout the intervention, suggesting that this type of exercise might mitigate the side effect of treatment. In particular, the increase in biological inflammatory markers level, such as interleukin-1β (IL-1β), IL-6, and tumor necrosis factor-α (TNF-α), is strongly related to the anthracycline treatment [132]. Moreover, Ansund et al. [133], studied the effect of HIIT training combined with Resistance or Aerobic exercises (RT+HIIT, AT+HIIT) on long-term myocardial damage in BC women undergoing anthracycline treatment. Eighty-eight BC women who started chemotherapy were randomized to 16 weeks of RT+HIIT, AT+HIIT, or usual care activity. RT+HIIT group performed resistance training, consisting of 8–12 repetitions at 75–80% of 1 RM, followed by 3 × 3 min bouts of aerobic HIIT on a cycle ergometer, whereas the AT+HIIT group started each session with 20 min of moderate-intensity continuous aerobic exercise, followed by the same HIIT protocol as RT+HIIT. Patients were assessed at baseline, at the end of the protocol (16 weeks), and one and two years after the end of training. After 16 weeks, all groups showed an increase in plasma cTnT level; the exercise groups, both RT+HIIT and AT+HIIT, maintained their cardiorespiratory fitness, which declined in the usual care group. The assessment at 1 year showed that Nt-pro-BNP was higher in the usual care group compared to RT+HIIT and AT+HIIT. It is interesting to note that those patients that reported a high level of cTnT and NT-proBNP during the first assessments showed a more pronounced decline in VO_2_peak after 2 years of follow-up. These data suggested that changes in these biomarkers’ level might single out compromised cardiovascular function in advance. HIIT either with resistance or aerobic training might provide a long-term cardioprotective effect in patients with breast cancer undergoing anthracycline. 

### 5.6. Exercise during Pandemic Restrictions

In the last two years, COVID-19 pandemic restrictions negatively affected the general population’s lifestyle, and the limited access to clinical practice increased sedentary behaviors in cancer patients with a great impact on health and quality of life [134,135]. Natalucci et al. evaluated the cardiometabolic effect of a home-based lifestyle intervention focused on the Mediterranean diet and aerobic exercise in BC survivors during the COVID-19 lockdown. Thirty BC patients performed aerobic training (between 40% and 70% of patients’ HRR) at least three times per week, and the specialized trainer supervised the activities using phone calls every week. At the end of 3 months, all patients significantly improved their VO_2_max, estimated through a submaximal test, while no changes were detected in systolic dysfunction (GLS > −18%). The number of patients who manifested signs of diastolic dysfunction decreased from 15 to 10 after the intervention. Moreover, cardiac function analysis showed a significant reduction in mean heart rate and an improvement in autonomic function after training. This study, conducted during the COVID-19 pandemic period, highlighted that a physical activity counseling program can induce positive effects on cardiometabolic health, decreasing the negative long-term side effects treatments in BC patients [136]. Moreover, the results suggest that this type of intervention is feasible even during an emergency period and could be an alternative strategy to encourage physical activity for these patients during confinement.

### 5.7. Suggestions for the Type of Exercise and the Exact Timing to Propose it to Breast Cancer Patients Undergoing Anthracycline

As most of the studies presented in this review evaluated the impact of the protocol during treatment, this timing might be the best with which to start a well-tailored exercise program. However, due to the paucity of scientific evidence, more studies are needed to confirm this suggestion. Regarding the type of exercise, aerobic training might be the most feasible and effective protocol, able to induce improvement in several functional, subclinical, and clinical parameters related to cardiotoxicity, such as VO_2_peak LVEF, BP [111,114]. Moreover, this type of exercise might attenuate the increase in those biomarkers that are prognostic for anthracycline-related cardiac injury and events [28,110]. Indeed, controversial results were found in BC patients that performed a combined training or HIIT. Studies on these protocols showed no impact on those clinical parameters related to cardiotoxicity, such as diastolic and systolic dysfunction [127,129], but cardiorespiratory fitness seemed to be maintained after treatment [133]. Further RCT is needed to understand if resistance, combined with aerobic training, may provide a long-term cardioprotective effect in patients with breast cancer undergoing anthracycline.
cancers-14-02288-t001_Table 1Table 1Effect of physical exercise on cardiotoxicity before, during, and after BC anthracycline treatment. Abbreviations: RC, Randomized Control Trial; NRCT, No-Randomized Control Trial; RPT, Randomized Prospective Trial; LS, Longitudinal Study; CSS, Cross Sectional Study; PS, Prospective Study; TR, Treatment; AT, Aerobic Training; RT, Resistance Training; CT, Combined Training; SAT, Single Aerobic Training; 4BAT, Four Bout Aerobic Training; HBAT, Home-based Aerobic Training; HIIT, High Intensity Interval Training; ST, Supervised Training; NST, NO-Supervised Training; CON, NO-exercise; w, week; d, days; min, minutes; s, seconds; rep repetitions; 1-RM, One Repetition Maximum; VO_2_max, Maximum Volume of Oxygen; VO_2_peak, Peak Oxygen Uptake; OP, Oxygen Pulse; PPO, Peak Power Output; QoL, Quality of Life; LBM, Lean Body Mass; FIT, Physically Active; UNFIT, NO-Physically Active; SVR, Systemic Vascular Resistance; BP, Blood Pressure; LVEF, Left Ventricular Ejection Fraction; IVRT, Isovolumetric Relaxation Time; DT, E peak deceleration time; HRR, Hearth Rate Reserve; RHR, Resting Hearth Rate; cTnT, Cardiac Troponin T; NT-proBNP, N-terminal-pro B-type natriuretic peptide; CK-MB, phosphokinase; Hb, Hemoglobin; HF, Hearth Failure; BC, Breast Cancer; W, watt; ↑, increase; ↓, decrease.AuthorTypeStudy PopulationInterventionOutcomesCourneya et al.(2007)[114]RCT(*n* = 242)Age = 25–78AT (*n* = 78)RT (*n* = 82)CON (*n* = 82)17 w, 3 d/w (During-After TR)AT: 45 min60–80% VO_2_max;RT: 3 d/w9 x 8-12 rep at60–70% 1 RM;AT: ↑ aerobic fitness, QoL, fatigue, depression, anxietyRT: ↑ LBM, QoL, self-esteem, depressionHornsby et al.(2014)[116]RCT(*n* = 20)Age 35–57AT *n* = 10CON *n* = 1012 w,3 d/w (During TR)AT: 60–100% VO_2_peak+1 d/w IT 10–15 x 30 sAt 100% VO_2_peak60 s Active RecoveryAT: ↑ VO_2_peak, OPNagy et al.(2017)[131]2 years-PS(*n* = 55)Age 31–65FIT *n* = 36UNFIT *n* = 19(During and After TR)FIT: more 30 min intensive exercise, 4–5 d/wUNFIT: less 30 min intensive exercises, 4–5 d/wFIT: ↓ Ea/Aa, HF symptomsUNFIT: ↓ Ea/AaKirkham et al.(2017)[110]RCT(*n* = 24)Age 40–60SAT *n* = 13CON *n* = 11(24 h Before TR)SAT: 30 min at 70% HRRSAT: ↑ NT-proBNP↓ SVR, diastolic BP, arterial BP↑ Stroke volume, cardiac output, LVEF, E/A, strainCON: ↑ NT-proBNP, end-diastolic volume, stroke volume, strain, twistKirkham et al.(2018)[28]RCT(*n* = 24)Age 40–604BAT *n* = 13CON *n* = 11(24 h Before TR)4BAT: 30 min at 70% HRR4BAT: ↑ cTnT, NT-proBNP↓ Depressed mood, sore muscles, low back painCON: ↑ cTnT, NT-proBNP, Cardiac output, RHR, Weight↓ Systemic Vascular Resistance, Arterial BPMa et al.(2018)[111]RCT(*n* = 70)Age 37–48AT *n* = 35CON *n* = 3516 w 3 d/w (During TR)AT: 50 minat 60–70% HRmaxAT: ↑ VO_2_max, LVEF, IVRTCON: ↓ VO_2_max, LVEF, E/A, DT interval↑ NT-proB-NP, CK-MBKirkham et al.(2019)[125]LS (*n* = 73)Age 29–77During TR: CT 3 d/w 20–30 min at 50–75% HRR and 1 RM + 1–2 d/w HBATImmediately after TR (10 w): CT 2 d/w + 3 d/w HBAT10 w after TR (10 w):1 d/w CT + 4 d/w HBAT VO_2_/HRR↑ HR rest↑ Aerobic Fitness↑ SBP rest, DBP rest, TR-related changesHowden et al. 2019[127]Prospective NRCT(*n* = 28)Age 33–64CT + HBAT *n* = 14CON *n* = 1412 w (During TR)2 d/w CT 60 min+ 1 d/w HBAT no supervised 30–60 minCT + HBAT: ↓ LVEF↑ Troponin ICON: ↓ VO_2_peak, PPO, LVEF↑ Troponin ILee et al.(2019a)[108]RPT(*n* = 29)Age 37–57HIIT *n* = 15CON *n* = 158 w, 3 d/w (During TR)HIIT: 1 min 90% PPO and 2 min 10% PPO for7 timesHIIT: = VO_2_max, PPOCON: ↓ VO_2_max, PPOLee et al.(2019b)[121]RPTsee Lee et al. (2019a)i.e., Lee et al. (2019a)HITT: ↑ baFMD, = cIMTCON: ↓ baFMD ↑ cIMTUpshaw et al.(2020)[126]3 years-LSAge 42–58(*n* = 603)(During and after TR)Godin Leisure-Time Exercise Questionnaire↑ baseline PA attenuates ↓ LVEFMoller et al.(2020)[118]RCT (*n* = 130)Age 42–61ST *n* = 64NST *n* = 6612 w (During TR)ST: 6 w, 9 h/w of CT+6 w, 6 h/w ofFloorball, Dance, and Circuit trainingNST: 150 min/w Moderate to Vigorous PASET and NSET: ↓ Metabolic Risk Profile= weight, VO_2_peakSET: ↑ muscle strength, LBM,1 RM knee extension, leg press, and lateral pullKirkham et al. (2020)[129]NRCT(*n* = 37)Age 40–60CT *n* = 26CON *n* = 113 d/w (During 4 cycles TR)CT: 20–30 min at 50–75% HRR + whole body resistance exercisesCT and CON: = Strain, VEF, E/A↓ Hb, hematocrit, and Arterial BPCT: ↓ Cardiac Output, VO_2_peakCON: ↑ Cardiac output↓ SVRLee et al.(2020)[122]RPT(*n* = 30)Age over 18HIIT (*n* = 15)CON (*n* = 15)8 w, 3 d/w (During TR)HIIT: 1 min 90% PPO and 2 min 10% PPO for7 timesHIIT: ↓ MMP-9CON: = MMP-9HIIT and CON: ↑ MMP-2Kirkham et al. (2021)[130]CSS(*n* = 54)Age 46–66FIT no BC *n* = 10FIT BC no TR *n* = 6FIT BC after TR *n* = 4UNFIT no BC *n* = 6UNFIT BC no TR *n* = 10UNFIT BC after TR *n* = 12(During TR)Godin Leisure-Time Exercise Questionnaire+Maximal Exercise TestFrom 20 W + 5 W every 20 s to volitional exhaustionBC after TR: ↓ VO_2_peak ↑ Myocardial FibrosisBC no TR and no BC: same VO_2_peak, Myocardial FibrosisAll FIT group: ↓ Myocardial FibrosisNatalucci et al.(2021)[136]RPT(*n* = 30)Age 30–70(After TR)AT: 3 w, 1 d/w NST, 2 d/w ST, 20–35 min at 40–50%HRRHBAT: 9 w, 3 d/w,20–60 min at 50–70% HRR+Weekly healthy lifestyle reminder↑ VO_2_max= GLS↓ diastolic dysfunction↓ mean heart rate↑ Autonomic functionAnsund et al.(2021)[133]RCT1- and 2-years FU(*n* = 88)Age 17–70RT-HIIT *n* = 29AT-HIIT *n* = 32CON *n* = 27(During TR)16 wRT-HIIT: 2 d/w 8–12 rep at 75–80% 1 RM + 3 × 3 min HIITAT-HIIT: 2 d/w 20 min + 3 × 3 min HIITafter 16 wRT-HIIT, AT-HIIT, and CON: ↑ cTnTRT-HIIT, AT-HIIT: = VO_2_maxCON: ↓ VO_2_max1 year FURT-HIIT, AT-HIIT: ↓ Nt-pro-BNPCON: ↑ Nt-pro-BNPHeinze-Milne et al. (2021)[132]PS(*n* = 49)Age 18–7012 w, 2 d/w (During TR)20–45 min ATat 35–85% HRR=VO_2_peak=IL-1β, IL-6, TNF-α, and VEGF

## 6. Conclusions

Due to the limited alternative strategies to reduce cardiac injury, the identification and testing of new interventions, pharmacologic and non-pharmacologic (or both), for preventing and/or treating anthracycline-induced cardiotoxicity is urgently needed. To date, cardiotoxicity is a frequent and devastating adverse complication of anthracycline therapy leading to morbidity, poor quality of life, and premature mortality. The evidence reviewed here indicates that aerobic exercise, even the HIIT protocol, might be a promising strategy to prevent and/or counteract chemotherapy-induced cardiotoxicity. The most relevant results were evidenced in those studies that introduced exercising during and not after the treatment, suggesting this timing to start a well-tailored physical activity as feasible and effective. Future studies are required to elucidate the molecular mechanisms underlying the cardioprotective properties of exercise before, during, and after anthracycline exposure, and how timing might influence these results. Collectively, hypothesis-driven translational studies are required to define the nature and magnitude of the cardioprotective effects of exercise in the setting of anthracycline chemotherapy. Such research will lead to mechanistically-driven clinical trials which, in turn, will introduce exercise into rehabilitation guidelines not only for patients affected by breast cancer, but also for patients with other solid anthracycline-sensitive malignancies.

## Figures and Tables

**Figure 3 cancers-14-02288-f003:**
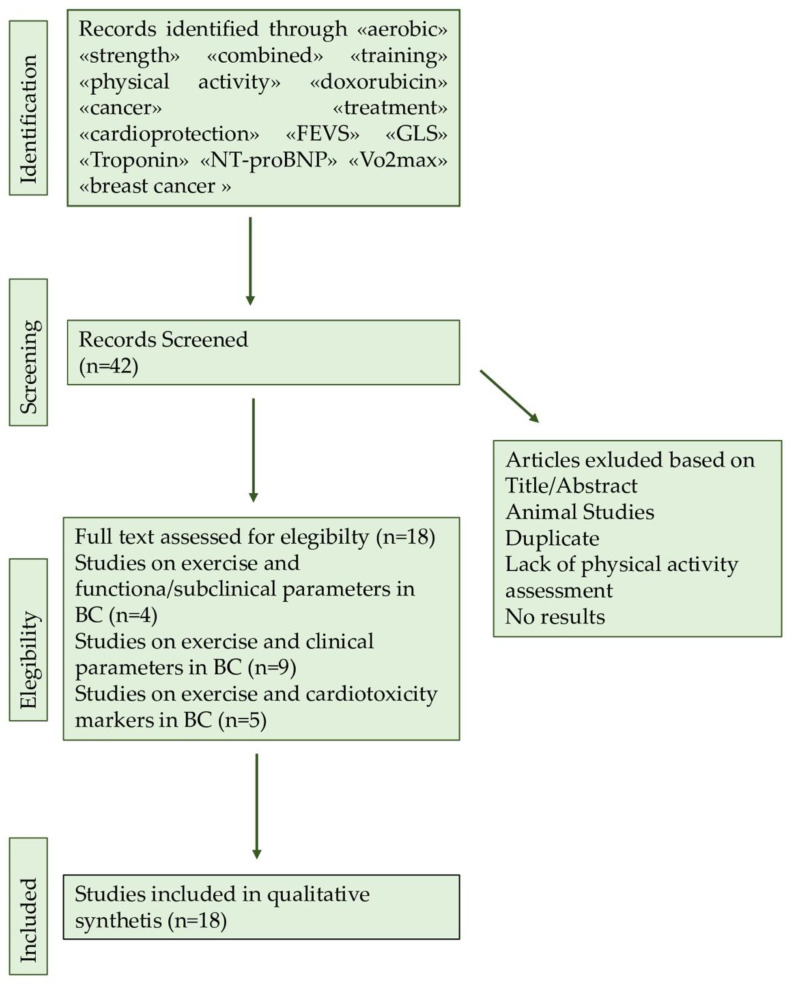
Flowchart of searching methods.

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
