# Peer review of "The Beneficial Role of Physical Exercise on Anthracyclines Induced Cardiotoxicity in Breast Cancer Patients"

_cancers, 2022, doi:10.3390/cancers14092288_

Round 1

Reviewer 1 Report

This is a review on the possible role of physical training in the prevention/control of anthracycline-induced heart disease. The topic is very relevant, and the aim to collect all the important data on and understand the mechanisms how anthracyclines cause heart changes from the molecular level to the clinical scenario is highly appreciated. The review follows a logical structure and contains a comprehensive set of data. Nevertheless, there are 2 shortages that hinders understanding and utility. First, in most of the text drafting is loose, hence accurate information relying on important details is missing. Second, English phrasing many times is not clear, even rereading can not help to understand the meaning of numerous sentences.

Detailed review:

Title: It is a problem all over the manuscript that the authors mention „chemotherapy-induced” cardiotoxicity, but in fact deal with anthracycxlines-induced cardiotoxicity only. This should be corrected consistently

Abstract: The abstract does not reflect the entire content of the review, it does not include any information on the molecular data and experimental results. Neither does it phrase the exciting outcome of the review notably that peri-chemotherapy physical exercise seems to have potential to prevent anthracycline-induced cardiotoxicity, and that special guidelines are still missing on how affected patients should be managed more efficiently.

Introduction: Important data are communicated but an important reference is missing (referring to the text in lines 59-60).

Section 2: This section starts with the description of the ways of action of anthracyclines, but then goes into those molecular changes that are discussed in the next section number 3. The text in lines 122-135 rather belongs to Section 3.

Figure 1: Oxydative stress and generation of free radicals seem to be more or less synonyms, probably both should not be used. In this figure, the inclusion of the possible pathological changes of the anatomical structures (discussed in the manuscript)  under the set of molecular factors contributing to cardiotoxicity would improve the information content.

Section 3: The text in 216-265 should be checked, and information on how the experimental set-up was arranged (as the authors point to the important features of type, duration, intensity and frequency of chemotherapy) in each referred paper should be provided, and especially the timing of the physical activity relative to the administration of chemotherapy since its prime interest.

Section 5: Since the authors do not provide and introduction or explanation to the meaning of those training-related parameters and other terms used, most of the readers would not gain information from this section. Please complete that. Too many details of unimportant studies (the mentioned studies have no actual relation to anthracycline-induced heart toxicity other than feasibility) are given while only few is about the effect of physical training during anthracycle-therapy and long-term follow-up. All the sub-sections seem exciting, but with limited understanding due to loose phrasing and extensive description with no conclusive statements. Subheading in lines 589-590 is not clear.

Author Response

We thank the Reviewer for the comments, here you can find attached our answers. As suggested, we edited the English language and style of the manuscript.

Reviewer 2 Report

Thank you for the opportunity to review this manuscript.

The topic of the review is highly relevant given the increasing awareness of cardiac toxicities related to oncologic treatment. Anthracyclines are the backbone of breast cancer treatment.  Eliana Tranchita et al have provided a comprehensive review of the pathophysiology of anthracycline induced cardiovascular toxicities. The manuscript followed the PRISMA protocol for systematic review. Studies are well summarized and all relavant references are cited. I commend authors for their effort highlighting current scientific evidence of role of different type of exercises and how to integrate it in future studies to find out how it may help in counteracting the cardiomyocyte injury from cytotoxic chemotherapies.

Few minor comments

The length of the article is a bit long. Can be shortened by rephrasing some of the sentences

Line 119: “through a direct inhibition of topoisomerase 2 in cancer cells”- replace it with topoisomerase II

Line 211 “ In 1979, it has been published the first study evaluating the role of exercise in chemotherapy-induced cardiotoxicity (CIC) in a mouse model “ – please correct the grammar  

Probably should be read “ the first study evaluating the role of exercise in chemotherapy-induced cardiotoxicity (CIC) in a mouse model was published in 1979

Author Response

(The authors gave the same response as above.)

Round 2

Reviewer 1 Report

I recommend this revised review paper to be accepted as it is now.